# To The Point: Correspondence-driven monocular 3D category reconstruction

**Filippos Kokkinos**
Department of Computer Science
University College London
`filippos.kokkinos@ucl.ac.uk`

**Iasonas Kokkinos**
Department of Computer Science
University College London, Snap Inc.
`i.kokkinos@cs.ucl.ac.uk`

## Abstract

We present To The Point (TTP), a method for reconstructing 3D objects from a single image using 2D to 3D correspondences learned from weak supervision. We recover a 3D shape from a 2D image by first regressing the 2D positions corresponding to the 3D template vertices and then jointly estimating a rigid camera transform and non-rigid template deformation that optimally explain the 2D positions through the 3D shape projection. By relying on 3D-2D correspondences we use a simple per-sample optimization problem to replace CNN-based regression of camera pose and non-rigid deformation and thereby obtain substantially more accurate 3D reconstructions. We treat this optimization as a differentiable layer and train the whole system in an end-to-end manner. We report systematic quantitative improvements on multiple categories and provide qualitative results comprising diverse shape, pose and texture prediction examples. Project website: `https://fkokkinos.github.io/to_the_point/`.

## 1  Introduction

Monocular 3D reconstruction of general categories is a task that humans perform with ease, yet remains challenging for computer vision due to its inherently ill-posed nature: the observed 2D image is the result of a confluence of multiple sources of variation, including non-rigid intra-category shape variation, rigid transforms due to camera pose, as well as appearance variation. CNNs can easily learn to discard appearance variation, yet the treatment of the geometric sources of variability remains elusive. Even though strongly-supervised approaches have delivered compelling results e.g. for human reconstruction [1], for general categories we need to rely on weaker forms of supervision as well as self-supervision stemming from the know-how of computer vision.

3D vision has traditionally relied on correspondences to recover both rigid scenes from 2D images for the Structure-from-Motion (SFM) problem [2, 3, 4] as well as the more challenging problem of recovering Non-Rigid structure from 2D point tracks (NR-SFM) [5, 6, 7, 8]. In all those problems 3D reconstruction is accomplished by minimizing the reprojection error between the 3D positions of the inferred 3D scene and their 2D image correspondences. While these solutions have been developed for the (potentially deformable) single-instance case, the idea of relying on correspondences to supervise monocular 3D reconstruction has transpired in recent deep learning works.

CNN-driven monocular 3D category reconstruction [9, 10, 11, 12] has largely relied on self-supervision for 3D recovery expressed in terms of correspondence-based loss terms. For instance the geometric cycle loss terms of [13, 14, 15] are explicitly phrased in terms of correspondence established from UV maps while the texture-driven loss terms of [9, 13, 12, 16, 11] are implicitly relying on pixel correspondence. The common ground of such loss terms is that if the 3D shape is predicted correctly, it should project to the image in a way that is consistent with the 2D observations, as measured in a pixel-by-pixel sense. These correspondence terms are typically used in

tandem with explicit geometric priors such as 3D symmetry [10, 9], predefined camera viewpoint ranges [14, 15, 10], or predefined object scales [14, 15, 10, 12] in order to tackle the ill-posed nature of the problem and the presence of multiple local minima in the associated learning problem.

Local minima however emerge even in the simpler single-instance case of NR-SFM, while highly sophisticated optimization schemes have been introduced to address them, e.g. [17]. Current CNN-based approaches seem to ignore this problem and further exacerbate it by delegating the solution of 3D reconstruction to back-propagation with SGD: separate network heads are tasked with regressing the camera pose and non-rigid deformation given an image and are trained in an end-to-end manner, aiming to minimize the correspondence-driven losses. We argue that this is making optimization harder: network training aims at simultaneously establishing the association between images and rigid and non-rigid pose parameters as well as solving the 3D reconstruction problem in terms of these parameters. Each of these problems is hard enough in isolation and putting them together makes the optimization even harder.

This challenge is reflected in the complicated numerical schemes currently used to mitigate local minima; for instance [10, 12] use multiple camera hypotheses during both training and testing. The number of hypotheses can range from 8 to up to 40 for a single reconstruction and the hypotheses have to be accompanied with a probabilistic method to select the most accurate pose either predicted by an MLP [14] or using heuristic loss-based weighting schemes [10]. Another example of brittle optimization, even when keypoint supervision is available, are the works of [9, 18] where in a first stage SFM/NR-SFM is used to get the camera pose right based on keypoint supervision, which is then followed by optimization with image-based losses to recover a mesh. This challenge has been observed also in the strongly-supervised case of human pose estimation, and the use of per-sample numerical optimization [19] was shown to improve performance in [20, 21, 22, 23, 24].

In this work we deviate from the current practice of using a CNN to regress camera and mesh deformation estimates. Instead, during both training and testing we solve a per-sample optimization problem that explicitly aims at providing a 3D reconstruction that projects "To The Point" (TTP). We take as input the 2D coordinates corresponding to the 3D vertices of a mesh and recover the 3D vertex positions by optimizing with respect to the rigid and non-rigid pose parameters through differentiable optimization [25, 26]. We obtain the 2D points required by our layer by only relying on mask annotations and optionally a small number of 2D semantic keypoint annotations, as well as self-supervision coming from the 3D reprojection loss. We jointly learn the 2D point regression and the 3D modes of shape variability through end-to-end optimization, while treating the per-instance rigid and non-rigid pose parameters as latent variables that are optimized on-the-fly, per sample.

We claim that predicting the correspondences is not only sufficient, but also more appropriate for driving monocular 3D reconstruction: it spares us from the use of any additional geometric priors and also yields state-of-the-art results while only relying on a single camera hypothesis. We evaluate our approach on 3D shape, pose and texture reconstruction on four objects categories using real-world datasets CUB [27] and PASCAL3D+ [28]. We demonstrate competitive 3D reconstruction quality to previous state-of-the-art methods and our ablation study confirms the importance of the self-supervised losses we employ.

## 2 Related Work

**Monocular 3D reconstruction** Recent works on this problem [29, 9, 14, 15, 16, 12, 30] have relied on varying forms of supervision. Earlier approaches [9, 29] treat the problem of 3D reconstruction from single images using known masks and manually labelled keypoints from single viewpoint image collections. Recent works [10, 16, 14] have removed the need for keypoints but introduced multiple viewpoint and deformation hypotheses accompanied with a probabilistic method to select the most accurate pose. [13, 31, 32] jointly recover cameras and non-rigid 3D meshes with single hypothesis-based networks, but limit themselves to simpler, almost planar categories like faces, or exploit symmetry priors, limiting their broader applicability.

Closer to our work is Canonical Surface Mapping (CSM) and Articulated Canonical Surface Mapping (ACSM) [14, 15] where the 3D representation is produced in the form of a rigid or articulated template using a 2D-to-3D cycle-consistency loss.

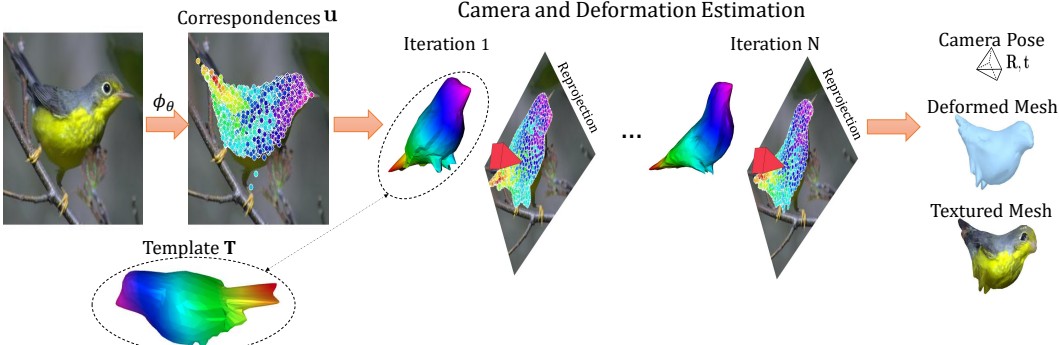

Figure 1: **Overview of our method:** Given an image we use a network $\phi_\theta$ to regress the 2D positions **u** corresponding to the 3D vertices of a template; we then use a differentiable optimization method to compute the rigid (camera) and non-rigid (mesh) pose: in every iteration we refine our camera and mesh pose estimate to minimize the reprojection error between **u** and the reprojected mesh (visualized on top of the input image). The end result is the monocular 3D reconstruction of the observed object, comprising the object's deformed shape, camera pose and texture.

**Non-rigid structure from motion (NR-SfM)** The aim of NR-SfM is the recovery of the 3D shape and accompanying camera pose given only 2D landmarks without any explicit 3D supervision [7, 8, 33, 34]. Lately, several deep learning [13, 35, 36] methods have been proposed that surpassed the performance of traditional methods while being considerably faster. All of the aforementioned methods employ different priors to tackle the under-constrained problem of NR-SfM. The priors are embedded into the methods using low-rank subspaces [37, 7, 17], spatio-temporal domains [34, 38], equivariance constraints [35] or sparse basis coefficients using L1 constraints [36, 39, 40].

**Learnable Optimization**: Common methods for incorporating optimization as layers in deep neural networks include implicit function differentiation [25, 26, 41, 42, 43] and optimization unrolling [44, 45, 36]; we refer to [26, 25] for a survey. In 3D reconstruction recent works address the challenge of incorporating RANSAC in an end-to-end trainable pipeline for camera pose estimation based on the Perspective-n-Point (PnP) problem, such as differentiable blind PnP [42, 43] or DSAC [46]. Unlike these works, we do not have to address the combinatorial nature of correspondence, but rather focus on regressing the 2D image positions of a 3D template with a fixed number of vertices.

## 3 To-The-Point Monocular 3D Mesh Reconstruction

We start in Sec. 3.1 by introducing the 2D quantities predicted by our network, we then present our differentiable camera and mesh optimization layer in Sec. 3.2, and finally present the losses driving our end-to-end training in Sec. 3.4.

### 3.1 Predicting 3D to 2D Correspondences

Our method assumes that the template of our object category can be described in 3D in terms of $N$ points. As shown in Fig. 1, given an image, we use a CNN, $\phi_\theta$, to regress the 2D coordinates $\mathbf{u} \in \mathbb{R}^{N \times 2}$ corresponding to these $N$ 3D points. We also predict a visibility vector **v** where every $v_i \in [0, 1]$ indicates whether the 2D to 3D correspondence is occluded in the image. Recognition of occluded points allows for accurate camera pose estimation by eliminating the influence of noisy predicted points belonging to the non-visible areas of the object.

### 3.2 Estimation of Pose and Deformation

Our aim is to estimate the camera pose and object deformation using only the predicted 2D points **u**, the visibility vector **v** and the template mesh **T**. We first introduce our assumptions about the rigid and non-rigid part of the shape and then turn to the resulting optimization problem.

Firstly, as in [9, 10, 14, 15, 16, 11], we model the 3D-to-2D projection through weak perspective [2]. This involves a $2 \times 3$ 3D-to-2D "scaled orthographic" projection matrix of the following form:

$$\mathbf{C} = \begin{bmatrix} s & 0 & 0 \\ 0 & s & 0 \end{bmatrix}, \tag{1}$$

where the scaling factor $s$ accounts for global scaling due to depth variation; given a set of 2D image points this is set to their standard deviation, yielding invariance of the pose parameters to similarity transforms of the 2D and 3D coordinate frames. The rigid pose parameters comprise a rotation $\mathbf{R}$ and translation $\mathbf{t}$ that account for viewing a 3D object $\mathbf{V}$ from a given camera position. The parametric estimates for the 2D projections of a 3D object can thus be obtained as follows:

$$\hat{\mathbf{u}}(\mathbf{R}, \mathbf{t}) = \mathbf{C}(\mathbf{RV} + \mathbf{t}) \tag{2}$$

where $\mathbf{V}$ is $3 \times N$, $\mathbf{u}$ is $2 \times N$, and we overload notation for $+$ assuming that $\mathbf{t}$ is replicated $N$ times.

Having covered the rigid object case, we now turn to the modeling of non-rigid categories. For this we rely on the deformable template paradigm [47] commonly used also in the NR-SFM literature [7, 8], and obtain a shape estimate $\mathbf{V}$ by adding offsets $\boldsymbol{\Delta}_V = \mathbf{Bc}$ to a template shape $\mathbf{T}$, yielding $\mathbf{V} = \boldsymbol{\Delta}_V + \mathbf{T}$. Combined with Eq. 2, we have the following parametric estimate of the 2D positions:

$$\hat{\mathbf{u}}(\mathbf{C}, \mathbf{R}, \mathbf{t}) = \mathbf{C}(\mathbf{R}(\mathbf{T} + \mathbf{Bc}) + \mathbf{t}) \tag{3}$$

which is bilinear in $\mathbf{R}, \mathbf{t}$ and $\mathbf{c}$.

So far we do not deviate substantially from recent works [9, 10, 16, 11] in terms of modelling: these also rely on the morphable model paradigm [47] and regress a shape update $\boldsymbol{\Delta}_V$ and a rotation matrix through network heads. A minor modelling difference is that we have a low-rank model for $\mathbf{Bc}$, while their shape update is driven by a high-dimensional latent vector. Our main difference is that rather than delegating to the network the task of predicting the 'right' values of $\mathbf{R}, \mathbf{t}, \mathbf{V}$, we directly optimize for them through a lightweight and differentiable optimization scheme by exploiting our regressed 2D correspondences. This 'optimizes out' these parameters and ensures our 3D inference will project as accurately as possible to the 2D points, rather than delegating the optimization to backprop and the network heads.

Our "To-The-Point" approach aims at minimizing the following re-projection error between the predicted 2D points $\mathbf{u}$ and the 2D point estimates deliver by the parametric, 3D-based prediction:

$$l(\mathbf{R}, \mathbf{t}, \mathbf{c}) = \sum_{i=1}^{N} v_i \|\mathbf{u}_i - \hat{\mathbf{u}}_i(\mathbf{C}, \mathbf{R}, \mathbf{t})\|_2^2 + \gamma \|\mathbf{c}\|_2^2, \tag{4}$$

where we weigh the discrepancy between the two quantities by the regressed visibility, ensuring that the reconstruction process is robust to occluded points. We also add a regularization weight $\gamma$ on the expansion coefficients to avoid instabilities in the first stages of training, when the basis $\mathbf{B}$ is still unknown and can lead to prematurely committing to large arbitrary deformations.

To minimize the loss term in Eq. (4), we use an alternating optimization scheme and perform separate updates for camera pose estimation and mesh deformation:

$$\hat{\mathbf{R}}^t, \hat{\mathbf{t}}^t \quad = \quad \underset{\mathbf{R}, \mathbf{t}}{\arg\min} \, l(\mathbf{R}, \mathbf{t}, \hat{\mathbf{c}}^t), \text{subject to} \, \mathbf{R} \in SO(3) \tag{5}$$

$$\hat{\mathbf{c}}^{t+1} \quad = \quad \underset{\mathbf{c}}{\arg\min} \, l(\hat{\mathbf{R}}^t, \hat{\mathbf{t}}^t, \mathbf{c}) \tag{6}$$

where at each step we use some of the previously estimated quantities (denoted by hat) as fixed and optimize with respect with the remaining ones to update their estimates.

Starting from the optimization in Eq. 5, we satisfy the constraint that $\mathbf{R} \in SO(3)$ by using the angle-axis representation $\mathbf{r}$ of the rotation $\mathbf{R_r}$ such that $\mathbf{R_r} = \exp[\mathbf{r}]_\times$ where $[\cdot]_\times$ is the skew symmetric operator . The underlying non-linear problem is solved using the L-BFGS optimizer [48] and when $t > 1$ is initialized with the estimate of the previous iteration. To backpropagate through L-BFGS we use the implicit function theorem as described in [42, 43].

Given the rotation matrix $\mathbf{R}$, we solve Eq 6 in closed form as follows:

$$\mathbf{c}^{t+1} = (\boldsymbol{\Omega}^\mathsf{T} \mathbf{X} \boldsymbol{\Omega} + \gamma \mathbf{I})^{-1} \boldsymbol{\Omega}^\mathsf{T} \mathbf{X} \boldsymbol{\Upsilon}. \tag{7}$$

Each row of $\mathbf{\Omega} \in \mathbb{R}^{2N \times K}$ is defined as $\mathbf{\Omega}_i = \mathbf{C}\mathbf{R}^{t+1}\mathbf{B}_i$ and $\mathbf{X} \in \mathbb{R}^{2N \times 2N}$ is a diagonal matrix containing the visibility vector $\mathbf{v}$. Finally, the vector $\mathbf{\Upsilon} \in \mathbb{R}^{2N}$ is defined as $\mathbf{\Upsilon} = \mathbf{u}_i - \mathbf{C}\mathbf{R}^{t+1}\mathbf{T}_i - \mathbf{C}\mathbf{t}^{t+1}$. For clarity we provide the analytic derivation of the closed form solution in the supplementary material. We also backpropagate through matrix inversion, meaning that the chaining of the update steps can be treated as a differentiable layer and be used in tandem with end-to-end training.

Even though the only quantities regressed by our CNN are the 2D positions and associated visibilities, the basis $\mathbf{B}$ that represents the shape variability of our category in Eq.3 is a parameter of this layer, is randomly initialized, and is estimated through back-propagation. As shown in the supplemental material, the basis elements learned this way can be intuitively understood, while our experimental results indicate that they suffice for the accurate recovery of intricate mesh deformations.

### 3.3 Texture

The final part of monocular 3D reconstruction is the estimation of the texture of the reconstructed 3D shape. The texture of the object is sampled from the input image utilizing the predicted 2D points $\mathbf{u}$ closely resembling the sampling-based texturing method of CMR [9]. Unlike CMR, we do not predict uv locations to sample pixel values for the image with a dedicated learnable regressor, but use the predicted 2D points $\mathbf{u}$ that drive our whole 3D reconstruction process. For this we use a sampling-based texture approach where the face color is computed by interpolating the $\mathbf{u}$ coordinates and then sampling from the texture map, i.e the input image in our case. Any losses applied on the estimated texture back-propagate information to the predicted correspondences allowing us to use appearance image cues in addition to foreground masks to enable accurate correspondence predictions.

### 3.4 Loss terms

To train our approach we incorporate different losses focusing on pose estimation, texture prediction as well as mesh regularization.

**Texture Loss** compares the rendered textured image $\tilde{I}$ and the image appearance in terms of the perceptual similarity metric of [49] after masking by the silhouette $S$:

$$L_{\text{pixel}} = \text{dist}\left(\tilde{I} \odot S, I \odot S\right).$$

We also apply the loss on the symmetric texture predictions by using a bilateral symmetric viewpoint and average the two viewpoints. The soft symmetry constraint ensures that the texture of the non-visible side is still inline with the visible side. This constraint has been employed in prior works [9, 16], however, unlike the proposed method it is commonly applied in a hard-coded manner by symmetrizing the texture across an axis.

**Points Chamfer Distance** enforces points to lie inside and cover the silhouette of the depicted object [29, 15]. In order to formulate our loss term we define $C^{\text{mask}}$ as the Chamfer distance field of the binary mask of silhouette $S$. Silhouette consistency simply enforces the predicted 2D correspondences of an instance to lie inside its silhouette. This can be achieved by penalizing the points projected outside the instance mask by their distance from the silhouette. Silhouette coverage enforces the predicted points $\mathbf{u}_i$ to fully cover the mask of the depicted object and allows us to predict better camera poses and mesh deformations.

$$L_{\text{Chamfer}} = \underbrace{\sum_i C^{\text{mask}}(\mathbf{u}_i)}_{\text{silhouette consistency}} + \underbrace{\sum_{p \in S} \min_{\mathbf{u}_i} \|\mathbf{u}_i - \mathbf{p}\|_2}_{\text{silhouette coverage}}.$$

**Region similarity loss** compares the object support computed from the mesh by a differentiable renderer [50] to instance segmentations $S$ provided either by manual annotations or pretrained CNNs using their absolute distance:

$$L_{\text{mask}} = \sum_i |S_i - f_{\text{render}}(V_i, \pi_i)|.$$

**Cycle and Visibility loss** Similarly to CSM [14] we use a cycle loss between the regressed 2D correspondences $\mathbf{u}$ and the projected 3D points to ensure that regressed points, that form neighborhoods in the template shape, remain close in the image space. The cycle loss is defined as

$L_{\text{cycle}} = \sum_i \|\mathbf{u}_i - \pi(\mathbf{V})\|_2^2$. Furthemore, visibility of correspondences aids the camera pose estimation in weakly supervised cases. The visibility loss encourages the predictor $\phi_\theta$ to encode the visible area of the mesh in an image by enforcing the predicted visibilities to be similar to those of the rendered z-buffer $\mathbf{v}^{\text{gt}}$

$$L_{\text{vis}} = \sum_i \left\| \mathbf{v}_i - \mathbf{v}_i^{\text{gt}} \right\|_1.$$

**Equivariance Loss** The point regressor should be robust to the pose variations. For each training image, we draw a random spatial transform $T_s(\cdot)$ from a predefined parameter range. We use random affine transformations (scale, rotation, and shifting) for spatial transforms as well as vertical flipping $T_v(\cdot)$. The detailed transform parameters are present in the supplementary material. We pass both the input image $I$ and transformed image $I' = T_s(I)$ through the $\phi_\theta$ network and obtain the corresponding predictions $\mathbf{u}$ and $\mathbf{u}'$. For vertical flipping we retrieve two pose estimations $\mathbf{R}$ and $\mathbf{R}'$ for $I$ and the flipped image $T_v(I)$. We compute the equivariance loss as follows:

$$L_{\text{eqv}} = \sum_i \|\mathbf{u}_i' - T_s(\mathbf{u}_i)\|_1 + \arccos \frac{1}{2}\left(\text{Tr}(T_v(\mathbf{R})\mathbf{R}') - 1\right).$$

**(Optional) Keypoint reprojection loss** While we are primarily interested in training without any manual annotations, our approach can be extended to leverage an arbitrary number of high-level semantic keypoints. This is achieved by setting manually the 3D keypoints on the template mesh and encoding them as a matrix $\mathbf{K}$ acting on the mesh. The structure of $\mathbf{K}$ entails that each $\mathbf{K}_i$ is a fixed vector that regresses the $i-$th 3D semantic keypoint from the mesh. Given the 2D annotations for an image $I$ and a camera $\pi$, a keypoint reprojection loss is formed between the groundtruth annotation and the projected 3D points:

$$L_{\text{kp}} = \sum_i \|\mathbf{k}_i - \pi(\mathbf{K}_i \mathbf{V})\|_1.$$

**As-rigid-as-possible (ARAP) constraint** Without any mesh deformation regularization, the predicted mesh deformation will lead to arbitrary deformations exhibiting spikes and other anomalies. As such, we use the as-rigid-as-possible (ARAP) [51] constraint as a loss function similar to [11]. The predicted shape $\mathbf{V}$ is a locally rigid transformation from the predicted base shape $\mathbf{T}$ by:

$$L_{\text{arap}}(\mathbf{T}, \mathbf{V}) = \frac{1}{N} \sum_{i=1}^{N} \sum_{j \in \mathcal{N}(i)} w_{ij} \left\| \left(\mathbf{V}^i - \mathbf{V}^j\right) - R_i \left(\mathbf{T}^i - \mathbf{T}^j\right) \right\|_2$$

where $\mathcal{N}(i)$ represents the neighboring vertices of a vertex $i$, $w_{ij}$ the cotangent weights and $R_i$ the best approximating rotation matrix, as described in [51]. Beyond mesh regularization, the same loss is applied on each basis component that leads to smooth and locally rigid components.

Even with ARAP there are cases where the network will squeeze the non-visible side of the reconstructed object. This erroneous deformation is not penalized by the ARAP loss, as long as it is locally rigid, and causes the method to predict flattened meshes. We further apply an $l_2$ constraint to the deformations to penalize the method to retain the original volume of the template.

## 4 Experiments

**Datasets** We present extensive ablation results and comparisons on bird reconstruction, as well as quantitative results on three more object categories (planes, cars, motorbikes). For birds we use the CUB [27] dataset for training and testing on birds which contains 6000 images. The train/val/test split we use for training and report is that of [9]. For the rest of the objects we use the Pascal3D+ dataset [28] and the associated pre-defined training and validation sets. Similarly to [9], we use both PASCAL VOC and Imagenet images to train our models and use Mask-RCNN [52] to obtain foreground masks for the ImageNet subset. For templates, we use identical to those of CSM [14] for CUB dataset and for PASCAL3D+ we select one of the available CAD models for each object.

**Evaluation Metrics** We evaluate our model on the CUB dataset [27] and report both the mean Intersection over Union (mIoU) and keypoint reprojection accuracy (PCK) following CMR [9].

Table 1: **Evaluation** of TTP performance on the CUB [27] dataset. We report mean and standard deviation (in parantheses, where applicable) of 2D mIoU and keypoint re-projection accuracy (PCK) along with related supervision signals for recent monocular 3D reconstruction methods.

| | 2D keypoints | Camera Priors | Camera Hypotheses | Rigid mIoU ↑ | Rigid PCK ↑ | Non-Rigid mIoU ↑ | Non-Rigid PCK ↑ |
|---|---|---|---|---|---|---|---|
| CMR [9] | ✓ | | 1 | - | - | 0.703 | 81.2 |
| (A)CSM [15] | ✓ | ✓ | 1 | 0.622 | 68.5 | 0.705 | 72.4 |
| ACMR [11] | ✓ | | 1 | - | - | 0.708 | 85.5 |
| TTP *(ours)* | ✓ | | 1 | **0.656** (0.002) | **70.0** (0.53) | **0.760** (0.004) | **93.4** (0.14) |
| (A)CSM [15] | | ✓ | 8 | 0.625 | **50.9** | 0.693 | 46.8 |
| (A)CSM [15] | | ✓ | 1 | 0.637 (0.004) | 39.0 (1.07) | 0.684 (0.011) | 44.5 (1.21) |
| TTP *(ours)* | | | 1 | **0.652** (0.008) | 48.7 (0.66) | **0.752** (0.003) | **50.9** (0.43) |

Table 2: **Performance** of TTP method through iterations for pose and deformation estimation. We achieve the best results with more iterations at inference, but even a single iteration suffices for competitive scores.

| | Number of Iterations 1 mIoU | 1 PCK | 2 mIoU | 2 PCK | 3 mIoU | 3 PCK | 4 mIoU | 4 PCK |
|---|---|---|---|---|---|---|---|---|
| TTP w/ KP | 0.732 | 92.5 | 0.755 | 93.3 | 0.758 | 93.3 | 0.758 | 93.4 |
| TTP w/o KP | 0.746 | 51.4 | 0.752 | 51.1 | 0.752 | 51.1 | 0.752 | 51.1 |

For Pascal3D+ we report a canonical 3D mean Intersection over Union metric which measures the 3D overlap between the groundtruth and predicted deformed mesh; in order to compute the overlap, both meshes are voxelized using a 32 grid size before computing the 3D mIoU as in [10, 9, 53, 29].

**Network Architecture** Following prior work [9], we use a ResNet18 encoder to map an image $I$ to a latent feature map $\mathbf{z} \in \mathbb{R}^{4 \times 4 \times 256}$. The position regressor is a fully connected layer having as input the flattened feature map $\mathbf{z}$ and outputs the regressed 2D positions $\mathbf{u} \in \mathbb{R}^{|V| \times 2}$ and their respective visibility $\mathbf{v} \in \mathbb{R}^{|V| \times 1}$. The number of basis components is set to $K = 16$ and the number of iterations for the camera and deformation estimation is four.

**Network Training** To train the 3D reconstruction model we first warm up the model without applying any deformation for 100 epochs. This warm-up process allows the model to find the best pose possible given the rigid template using available cues like masks, texture and optional keypoints. We then train the full 3D reconstruction network with deformation enabled and all available cues for another 100 epochs. All training details can be found in the supplementary material. All experiments were run on a single RTX 2080 Ti GPU.

## 4.1 Quantitative Results

**Evaluation on CUB** In Table 1 we evaluate TTP on the CUB dataset and report the average and standard deviation of 5 experiments with different seeds; the rigid part of the results table indicates the performance of models that do not use a deformable component, while the non-rigid part amounts to the more challenging problem of estimating both camera and mesh deformations.

We observe that our method outperforms the baseline models on both reported metrics, i.e. mean IoU and keypoint re-projection accuracy, while requiring no camera priors. When using 2D keypoint supervision (upper part of the table) our method achieves the best results, outperforming the closest baseline by almost 8 accuracy points. For the case where no 2D keypoints are used the only published result in the literature is the ACSM approach of [15] which relies on 8 camera hypotheses during both training and testing, while also using manual annotation of part-based rigs to bootstrap the deformation model. Our work outperforms this baseline while relying on a single camera hypothesis and without requiring any manual mesh annotation.

Table 3: **Ablation Study** We ablate the self-supervised losses used to train TTP for 3D reconstruction for the case of training both with and without keypoint supervision. We also study the impact of the number of basis elements.

(a) **Ablation on losses**.

|  | With KP | | Without KP | |
|---|---|---|---|---|
|  | mIoU ↑ | PCK ↑ | mIoU ↑ | PCK ↑ |
| TTP | 0.765 | 93.6 | 0.749 | 50.9 |
| TTP - $L_{pixel}$ | 0.752 | 92.7 | 0.667 | 49.2 |
| TTP - $L_{vis}$ | 0.75 | 92.3 | 0.74 | 9.3 |
| TTP - $L_{equiv}$ | 0.751 | 92.5 | 0.71 | 28.2 |

(b) **Ablation on number of basis components**.

|  | With KP | | Without KP | |
|---|---|---|---|---|
| Basis | mIoU ↑ | PCK ↑ | mIoU ↑ | PCK ↑ |
| rigid | 0.657 | 70.7 | 0.646 | 48.4 |
| 4 | 0.726 | 88.2 | 0.72 | 47.9 |
| 8 | 0.745 | 90.6 | 0.733 | 50.4 |
| 16 | 0.765 | 93.6 | 0.749 | 50.9 |
| 32 | 0.771 | 93.7 | 0.748 | 49.8 |
| 64 | 0.775 | 94.2 | 0.752 | 49.8 |

Table 4: **PASCAL3D+ evaluation**. We provide numerical score of TTP with and without keypoint supervision during training. We observe that *even without keypoint supervision* TTP is competitive with the other methods which, except for UCMR, require keypoints.

|  | CSDM [29] | DRC [53] | UCMR [10] | CMR [9] | TTP w/ KP | TTP w/o KP |
|---|---|---|---|---|---|---|
| aeroplane | 0.4 | 0.42 | - | 0.468 | **0.488** | 0.45 |
| car | 0.6 | 0.67 | 0.646 | 0.64 | **0.67** | 0.665 |

We posit that using multiple cameras in [15] aims at mitigating the local minima in network training and optimization. We have therefore rerun the system of [15] with a single camera and five different optimization seeds and observed a further gap in performance compared to our work, as well as a larger variance in the reconstruction accuracy compared to that of our work for the non-rigid case, suggesting a potentially higher chance of getting stuck in local minima.

**Ablation study** We ablate various terms in our learning objective and report the mIoU and the semantic keypoint reprojection (PCK) metrics. In particular we examine the impact of removing any of the utilized losses in Table 3a. When using keypoint supervision the differences in performance are small. However in the absence of keypoints, the method struggles to align the template with the depicted object when we remove the visibility loss. While mIoU remains high, PCK score decreases substantially meaning that the pose and deformation of the template cover the foreground mask when rendered but not from a proper viewpoint of the object. Similar performance drop occurs without the equivariance loss since the method produces pose estimates biased towards one vertical direction. Finally, removing the texture loss causes mIoU performance to drop significantly.

In Table 3b we study how performance changes as a function of the number of basis elements. Increasing the number of components tends to increase performance but up to 16 elements for both set of experiments. When training with semantic keypoints increasing the number of basis components further improves performance, however since the same does not apply to the mask-only case we have set $K = 16$ for all of our experiments.

Finally, a key aspect of TTP is the iterative pose and deformation estimation process. In Table 2, we provide the mIoU and PCK scores for every iteration for two experiments trained with and without keypoint supervision. Multiple iterations have to be executed to get the best performance, however TTP's performance with a single iteration still outperforms prior work for both metrics.

We are complementing these quantitative results with qualitative results in Figure 2 where we show that we can correctly deform the template mesh to produce highly accurate 3D reconstructions.

**Evaluation on Pascal3D+** While our primary evaluation is on the CUB dataset, we run supplementary experiments on the cars, airplanes and motorcycle categories of PASCAL3D+ dataset. For cars and aeroplanes we provide comparisons against CMR [9], UCMR [10], a volumetric prediction network [53] and a fitting based method [29]. Three of the methods use segmentation masks, cameras and keypoints for supervision except UCMR that does not require keypoints.

We train our method with and without keypoint supervision and provide our 3D mIoU results in Table 4. We observe that our method performs considerably better than competing methods even when

CMR UCMR Ours-**u** Ours-Mesh Ours-Texture New view

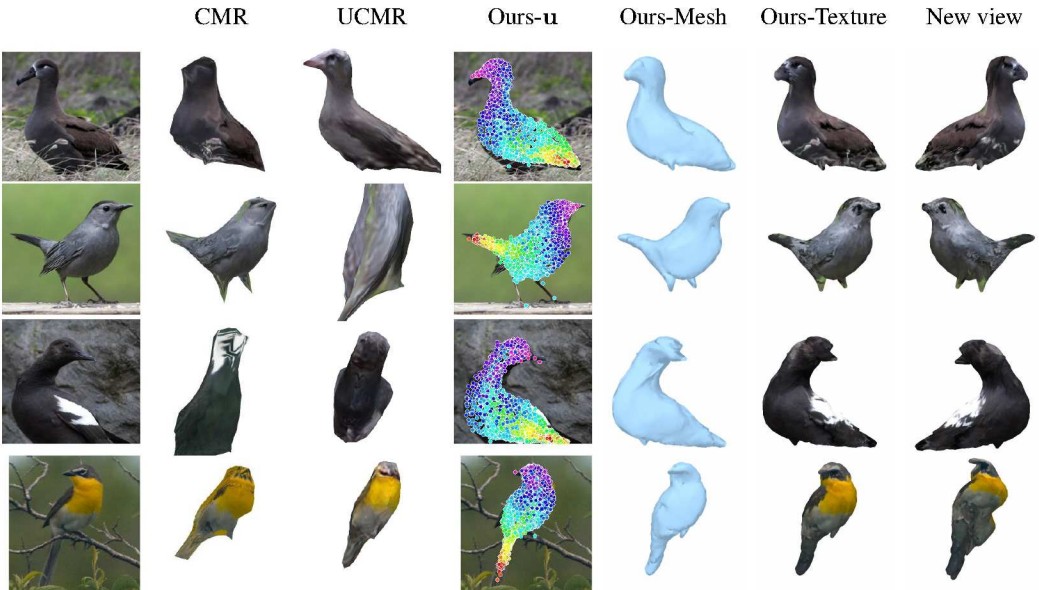

Figure 2: **Bird reconstructions** For each input image we provide the results of CMR [9] and UCMR [15] alongside with our method. We visualize the input image, predictions from prior works and TTP's predicted correspondence (**u**), mesh reconstruction and textured mesh from two viewpoints. We observe that we better capture texture details, deformation and pose estimation.

Mesh Texture New view Mesh Texture New view

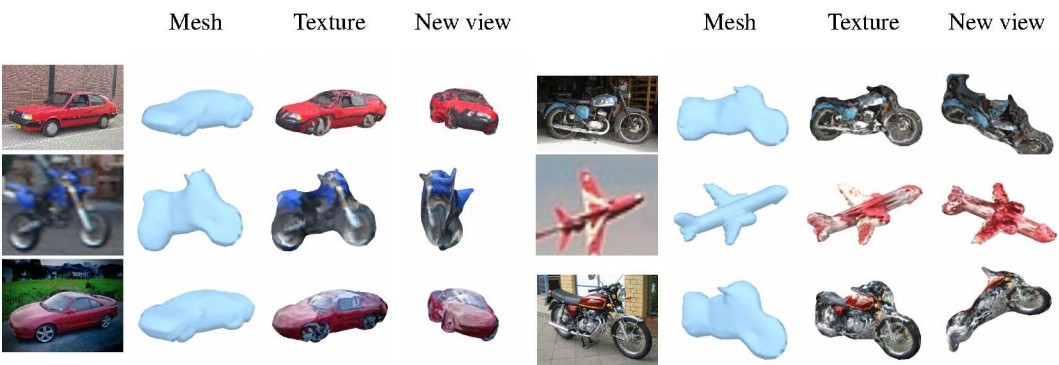

Figure 3: **Pascal3D+ results** We show predictions of our TTP method for test set images. For each input image we visualize the 3D shape from the predicted camera, the textured shape and a new view.

keypoint supervision is not utilized. Beyond the 3D mIoU metric we also provide the PCK scores of our method for the cars dataset and compare it against the only available reported method [14]. Our approach trained with and without keypoints results in 74.9 mIoU and 45.7 PCK scores while CSM achieves 51.2 and 40.0 respectively. The difference is significant in both cases, especially in the keypoint-free methods where predicting a correct camera pose is challenging due to the weak supervision setup. We provide qualitative results of TTP in Figure 3 and the supplementary material.

## 4.2 Failure Cases

We visualize some failure cases of the proposed method in Figure 4. Common failure cases are related to the inability to predict a good camera pose and 2D points.

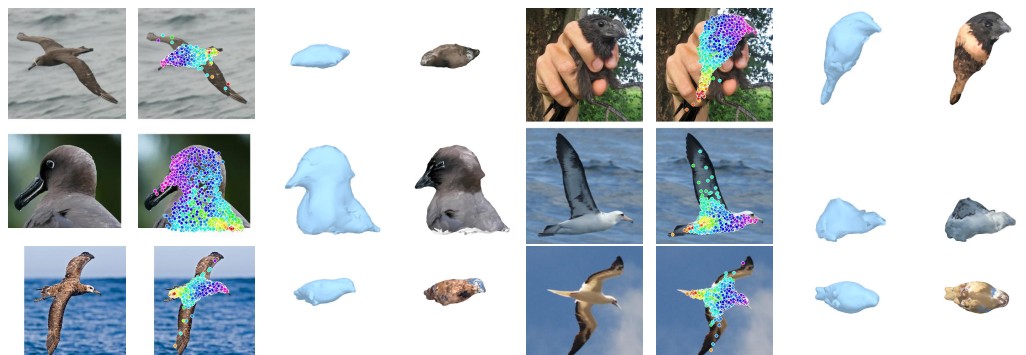

Figure 4: **Failure Cases:** We visualize some failure modes of our method. The columns present the input image, the predicted 2D points, and 3D reconstruction with and without texture.

Table 5: **Run-time analysis** in milliseconds of various self-supervised 3D methods. All benchmarks were run 20 times using images of size 256x256 and we report the average run times.

|      | Device | Iters | CNN (msec) | Optimization (msec) | Total (msec) |
|------|--------|-------|------------|---------------------|--------------|
| CMR  | GPU    | -     | 5.11       | N/A                 | 5.11         |
| CSM  | GPU    | -     | 150.73     | N/A                 | 150.73       |
| ACSM | GPU    | -     | 191.46     | N/A                 | 191.46       |
| TTP  | GPU    | 1     | 4.02       | 33.44               | 37.46        |
|      |        | 4     | 4.02       | 125.14              | 129.16       |
| TTP  | CPU    | 1     | 18.52      | 33.44               | 51.96        |
|      |        | 4     | 18.52      | 125.14              | 143.67       |

## 4.3 Run-time Analysis

In Table 5 we provide a run time analysis of our and prior methods on the task of self-supervised 3D reconstruction. The analysis was performed on a machine with a NVidia RTX 2080Ti, an Intel Xeon W-2255 and 32 GBs of RAM. We report the average of 20 runs for the reconstruction of a 256x256 image. To ensure that the benchmark is fair for all methods, we compute the execution time between the moment an image is given as input to a network up to the moment where the network predicts the mesh, the camera pose and the texture; the implementations of the methods used for comparison are the publicly available ones from the original authors. As reported in Table 5, TTP is faster than ACSM by a considerable margin even on the CPU. CMR is the fastest method of all due to its simplicity, however, it requires keypoint supervision and has substantially lower results as indicate in Table 1.

We note that our implementation relies on PyTorch and we have used PyTorch's python-based LBFGS optimizer for convenience and autograd for Jacobian computation; understandably for AR applications the LBFGS timing can become substantially faster in C or custom CUDA kernels and with explicitly coded Jacobian matrix computation.

## 5 Discussion

We have proposed a method to reconstruct 3D meshes, poses and textures of generic objects in the wild without any direct supervision. We learn unsupervised correspondences between 2D image locations and 3D template vertices and use them to compute the camera pose and deformation of the object. Even though our CNN architecture predicts substantially fewer outputs - compared e.g. to [9] where all of the 3D vertices and the camera are directly regressed by the network, we deliver substantially better results. We attribute this to the use of a direct optimization scheme to optimize the 3D reconstruction problem both during training and testing. The resulting optimization problem is particularly lightweight, meaning that it can be used for interactive applications, e.g. in Augmented Reality, while our results indicate that even a single step of the optimization suffices for accurate mesh recover. In future work we aim to extend our approach to cover categories with diverse topologies (e.g. chairs) as well as exploit video-based supervision [12] to further improve accuracy.

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
