## A   Appendix

We provide additional results including visualizations of the learned deformations, comparisons to CMR [9] and UCMR [10], qualitative results on a collection of different objects and technical details about the training procedure.

## B   Method details

**Architecture** We use the encoder-decoder architecture presented in [9, 10]. Every image is encoded using an ImageNet pre-trained ResNet18 to a latent feature map $z \in \mathbf{R}^{4 \times 4 \times 256}$. A flattened version of $z$ is processed with one linear layer with output channels equal to $N \times 3$ to get the predictions for points $\mathbf{u}$ and visibility $\mathbf{v}$. We apply the sigmoid function to the visibility predictions $\mathbf{v}$ to enforce a numerical range [0,1]. Our models are trained using Adam optimizer with learning rate equal to 1e-4.

**Equivariance** Equivariance loss samples random affine transformations $T_s(\cdot)$ from a predefined range. In detail, scale is sampled from the range [0.7, 1.2], vertical translation is up to 38 pixels and we also apply 2D rotation up to 40 degrees. For camera equivariance the image is simply flipped horizontally and given as input to the network to estimate the pose. We set the camera equivariance loss weight to 0.1 to ensure similar numerical range with the 2D point equivariance component.

**Implementation** We implemented TTP using PyTorch framework. All models were trained using a single NVidia 2080Ti with 12 GB of memory and a full experiment requires around 48 hours of computational time. TTP takes one average 0.149 seconds to reconstruct a single image using 4 iterations and 0.071 seconds using one iteration.

## C   Closed Form Solution of Quadratic Deformation Problem

In this section we provide the closed form solution of the deformation step. The problem we solve is

$$
\begin{aligned}
l(\mathbf{c}) &= \sum_{i=1}^{N} v_i \left\| \mathbf{u}_i - \mathbf{C}(\mathbf{R}(\mathbf{T}_i + \mathbf{B}_i\mathbf{c}) + \mathbf{t}) \right\|_2^2 + \gamma \left\| \mathbf{c} \right\|_2^2 \\
&= \sum_{i=1}^{N} v_i \left\| \underbrace{(\mathbf{u}_i - \mathbf{C}\mathbf{R}\mathbf{T}_i - \mathbf{C}\mathbf{t})}_{\mathbf{y}_i} - \underbrace{\mathbf{C}\mathbf{R}^{t+1}\mathbf{B}_i}_{\boldsymbol{\omega}_i} \mathbf{c} \right\|_2^2 + \gamma \left\| \mathbf{c} \right\|_2^2
\end{aligned}
\tag{8}
$$

where $\mathbf{y} \in \mathbb{R}^2$ and $\boldsymbol{\omega} \in \mathbb{R}^{2 \times K}$. The stationary point of (8) can be found by solving the following linear system:

$$
\begin{aligned}
&\sum_{i=1}^{N} v_i(-2\boldsymbol{\omega}_i^\mathsf{T}\mathbf{y}_i + 2\boldsymbol{\omega}_i^\mathsf{T}\boldsymbol{\omega}_i\mathbf{c}) + \gamma\mathbf{I} = 0 \\
&\sum_{i=1}^{N} (-2\boldsymbol{\omega}_i^\mathsf{T} v_i\mathbf{y}_i + 2\boldsymbol{\omega}_i^\mathsf{T} v_i\boldsymbol{\omega}_i\mathbf{c}) + \gamma\mathbf{I} = 0
\end{aligned}
\tag{9}
$$

The closed form solution can be further simplified using matrix representations. Assuming a lexicographic ordering of matrix $\boldsymbol{\omega}_i$ we formulate the matrix $\boldsymbol{\Omega}$ with size $2N \times K$. The matrix $\mathbf{X} \in \mathbb{R}^{2N \times 2N}$ is a diagonal matrix where each element of the diagonal corresponds to the visibility $v_i$. A lexicographic ordering is also applied to vectors $\mathbf{y}_i$ to retrieve the stacked vector $\boldsymbol{\Upsilon} \in \mathbb{R}^{2N}$. Using the formulated matrices the problem is solved with

$$
\mathbf{c}^{t+1} = (\boldsymbol{\Omega}^\mathsf{T}\mathbf{X}\boldsymbol{\Omega} + \gamma\mathbf{I})^{-1}\boldsymbol{\Omega}^\mathsf{T}\mathbf{X}\boldsymbol{\Upsilon}.
\tag{10}
$$

Backpropagation through the matrix inversion of Eq. (10) is supported by all modern automatic differentiation frameworks.

# D More results

## D.1 Comparisons on CUB

In Figure 5 we provide comparisons against prior work on common training supervision. It is apparent that our method is capable of correctly deforming the template mesh to produce bending and turning the body and head of the birds. In nearly all results, UCMR lacks a texture prediction that follows closely the depicted object for example in Row 5 and 6. Our method produces accurate 3D reconstructions both in terms of mesh deformation, pose estimation and texture.

In Figure 6 we provide comparisons against UCMR [10] and TTP trained without keypoints. All results of UCMR were computed using the publicly available code and pre-trained models for birds. It is visible that UCMR will often fail to predict either a correct scale or a rotation even in trivial input images like first column of Row 1. On the other hand, our method accurately estimates camera pose due to the proposed framework of computing jointly the camera pose and deformation using correspondences. In Figure 7 we provide comparisons of TTP where one model is trained with keypoints while the other is not. Semantic keypoints allow for more diverse deformations, however both methods will accurately predict the underlying pose.

## D.2 Basis visualization

We visualize some basis components of a trained TTP method in Figure 8 from a side and a top viewpoint. The basis components exhibit interesting deformations like head or tail bending and enlargement. The linear weighted combination of all the components is the building stone of our method to achieve accurate mesh deformations.

CMR  UCMR  Ours-**u**  Ours-Mesh  Ours-Texture  New view

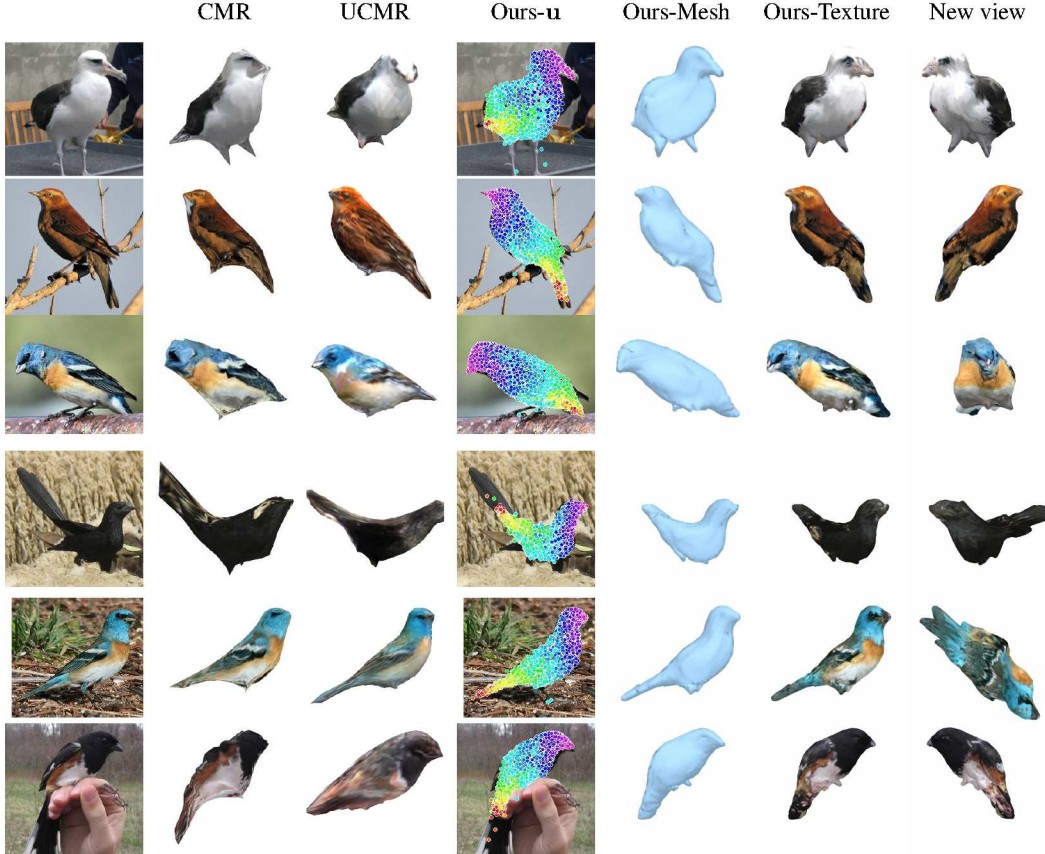

Figure 5: **3D reconstructions** For each input image we provide the results of CMR [9] and UCMR [15] alongside with our method. We visualize the input image, predictions from prior works and TTP's predicted correspondence (**u**), mesh reconstruction and textured mesh from two viewpoints. We observe that we better capture texture details, deformation and pose estimation.

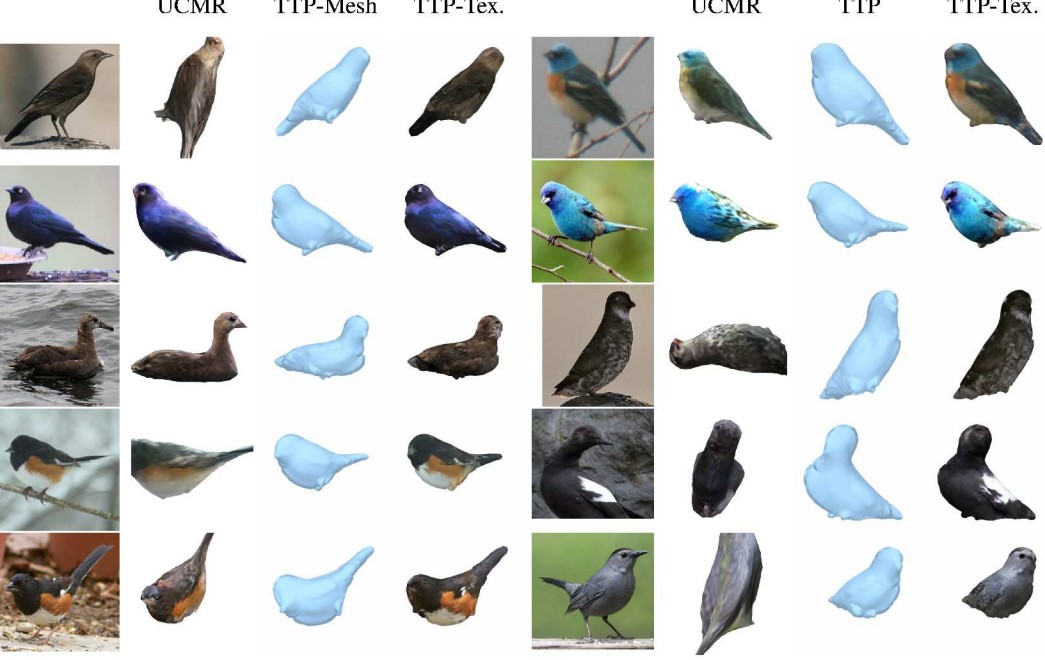

Figure 6: **No-keypoint supervision comparisons**. For each input image we provide the results of UCMR [10] alongside with our method. We visualize the input image, predictions from prior works and TTP's predicted mesh reconstruction and textured mesh. We observe that we better capture texture details and pose estimation.

TTP$_{kp}$ **u**  TTP$_{kp}$ 3D  TTP$_{kp}$ Tex.  TTP$_{nokp}$ **u**  TTP$_{nokp}$ 3D  TTP$_{nokp}$ Tex.

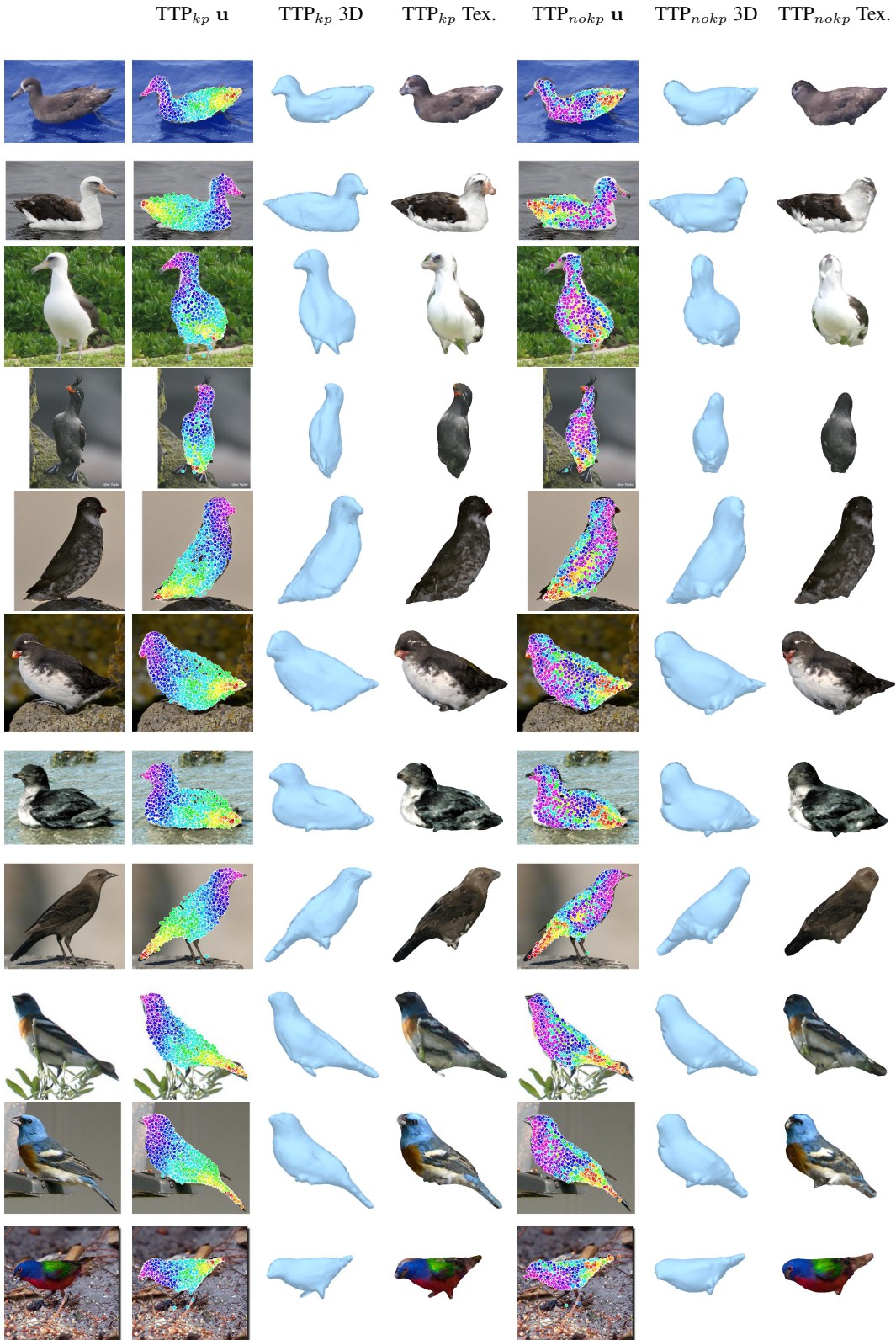

Figure 7: **Comparison of TTP trained with and without keypoint supervision**. We observe that TTP performs equally well on camera pose estimation with and without keypoint supervision. However, keypoint supervision allows for more accurate mesh deformation.

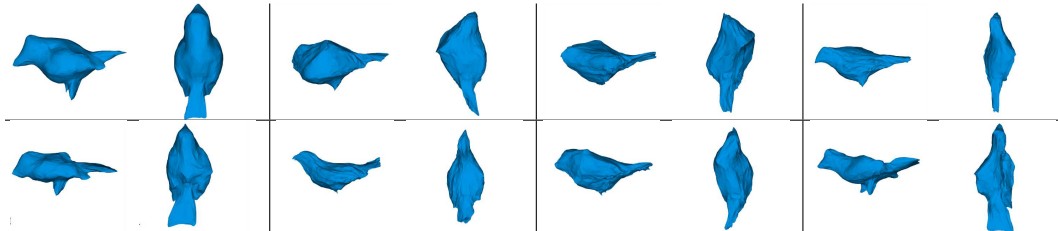

Figure 8: **Basis Visualization:** We visualize basis components $T + B_i$ of a trained TTP experiment for CUB dataset. For each component we visualize a side and a top view.

| Mesh | Texture | New view | | Mesh | Texture | New view |
|------|---------|----------|--|------|---------|----------|

Figure 9: **Pascal3D+ results** We show predictions of our TTP method for test set images. For each input image we visualize the 3D shape from the predicted camera, the textured shape and a new view. The last row represent failure case where the texture of non-visible side exhibits artifacts.