# OpenReview forum: "To The Point: Correspondence-driven monocular 3D category reconstruction"
_NeurIPS.cc/2021/Conference — NeurIPS 2021 Poster_

### Official Review · Reviewer_P1yY · 2021-07-07

**Rating:** 8
**Confidence:** 4

**Summary:**

The paper tackles the problem of 3D shape texture and pose estimation from a single image without 3D, multi-view nor camera calibrations. This is an extremely challenging problem. Concurrent works are CMR, U-CMR, CSM and A-CSM. **The key novelty** is to avoid learning pose, texture and shape with several networks heads. Instead, the author propose to learn the 2D positions corresponding to the 3D template vertices, then solve a per-sample optimisation to estimate the pose and template deformation, enabling texture recovery through differentiable rendering. They use implicit differentiation to differentiate through the L-BFGS solver of their inner PnP problem. They achieve successful reconstruction of cars, birds and motorcycles. They outperform their competitors on the Cub dataset and Pascal3D+ dataset, showing exciting successes even without keypoint supervision.


**Limitations And Societal Impact:**

# Weaknesses

-   A few things are still unclear to me. How are the unseen parts of the object textured? Using the predicted **u** ? Does this make sense considering they are not not supervised by the texture loss due the the visibility term? You may have discussed this in this sentence that I did not understand: “We also apply the loss on the symmetric texture predictions by using a bilateral symmetric viewpoint and average the two viewpoints. The soft symmetry constraint ensures that the texture of the non-visible side is still inline with the visible side.”
-   The current limitations of the method are not very clear. Is it easy/hard/impossible to train on a new category like humans? What is the current bottleneck limiting the quality of the reconstructions? Data, deformation model, convergence… ?



**Main Review:**


# Writing

The method is complicated, and the high quality of the writing helps communicate hard concepts.
How the method relates to the state-of-the-art is not really discussed  in the related work section. I am not sure "Non-rigid structure from motion (NR-SfM)" deserves a section.




# Strengths

The method is simpler than U-CMR and sets the new state of the art in shape, texture, and pose estimation from a single image. Avoiding to learn things that can be optimised per-sample seems like a more robust and simple approach.
I found the approach very original and am excited by the adoption of implicit differentiation in 3D deep learning. I greatly enjoyed the paper.

 # Post-rebuttal comment

I have read the other reviews and thank the author for their response. I found this approach very original and maintain my review.

**Time Spent Reviewing:**

7 hours

---

> ### Author Response · Authors · 2021-08-10
> **Response to Reviewer's feedback**
>
> *Comment: How the method relates to the state-of-the-art is not really discussed in the related work section. I am not sure "Non-rigid structure from motion (NR-SfM)" deserves a section.*
>
> We discussed the relation of our work with relevant state-of-the-art methods in the Introduction instead of the Related Work section. TTP draws inspiration from classic non-rigid structure from motion techniques as such we thought it would be useful for unfamiliar readers to discuss them in a separate section.
>
>
> *Comment: How are the unseen parts of the object textured? Using the predicted u ?  ... is still inline with the visible side.”*
>
> The unseen parts of the object are textured using the predicted u, but the predicted visibility term doesn't affect the texture loss. The predicted texture of the invisible side is learned using a symmetric assumption which means that the depicted object exhibits textural symmetry across left-right. This allow us to render the invisible side of the mesh with a simple camera transformation (left-right flip) and apply the same texture loss for the flipped rendering. Please note that this technique has been used by prior work [10] and doesn't constitute our contribution.
>
>
> *Comment: The current limitations of the method are not very clear ... deformation model, convergence… ?*
>
> The proposed method could be used potentially  for categories that exhibit large articulations like humans. However, in the current setup we use a linear deformation basis which is not necessarily ideal for human motion or hands. A step towards that direction will be to incorporate a parametric model like SMPL or MANO that have been learned using 3D scans. Further, a self-supervised setup will not suffice for these classes and 3D datasets for direct supervision is needed as it is common in most human/hand reconstruction works [1, 19, 20, 21, 22, 23, 30, 39]. One more limitation stems from the dependence on templates which limits the variance of the predicted deformations. Feature work should focus on addressing this limitation by employing a different geometric representation like SDFs. We will revise our manuscript to address better the current limitations of the method.

---

### Official Review · Reviewer_YB2E · 2021-07-11

**Rating:** 6
**Confidence:** 4

**Summary:**

The paper proposes a method for category-level 3D mesh reconstruction from single image. The key innovation is a correspondence-driven optimization layer that produces camera pose and object deformation given a single image, as opposed to using conventional pose CNNs. Overall, the idea is sensible, and the results are better than prior methods under the same setup, either given keypoint annotation or without keypoint annotation. My concerns is mostly on significance and writing.

**Limitations And Societal Impact:**

Yes

**Main Review:**

Originality and significance:
- The problem is well-motivated and the discussion of bad local minima in NR-SFM is helpful for understanding the significance of the problem.
- The proposed solution of casting the pose estimation problem as correspondence regression + optimization is also sensible. I can see how the extra optimization step could produce more accurate shape at test time. However, it's not obvious why it will address the bad local minima issue at training time. It would help to provide further insights or experiment evidence.

Clarity:
- The presentation is clear and easy to follow in general.

Method:
- The method section is well written in general. However, some details are missing from Sec 3.2 and makes it difficult to evaluate the technical significance. For instance, l136 the authors mentioned the optimization is lightweight, but without mentioning the complexity of each iteration and number of iterations needed for convergence.  In table2, the number of iterations is reported but it's not clear whether this is at test time or training time. It will help to compare to prior CNN-based methods in terms of accuracy and speed.
- From an input image, TTP first predicts 2D-to-3D-template correspondence, and then solves a nonrigid perspective-n-points problem that produces camera pose and shape deformation. However, I'm able to find a concrete description of how is correspondence trained. Is it fair to say the correspondence is trained by cycle consistency loss L_cycle?
- It is not clear how much deformation can the proposed approach handle. In Eq (7) the authors models shape deformation with a linear shape basis, that has a closed-form solution. However, a linear basis is likely to be insufficient for modeling large motion, such as articulation of human bodies.


Minor
- l106: I'm not following why "3D to 2D". From the description, it's the other way around.
- l142: weigh->weight

**Time Spent Reviewing:**

3

---

> ### Author Response · Authors · 2021-08-10
> **Response to Reviewer's feedback**
>
> *Comment: The method section is well written in general. However, some details are missing from Sec 3.2 and makes it difficult to evaluate the technical significance. For instance, l136 the authors mentioned the optimization is lightweight, but without mentioning the complexity of each iteration and number of iterations needed for convergence. It will help to compare to prior CNN-based methods in terms of accuracy and speed.*
>
> We provided a benchmark in the supplementary material, but we understand we should have elaborated more on the run time analysis. As such we refer the reviewer to the relevant answer for  Reviewer 1. The number of maximum iterations for L-BFGS is set to 100
>
> *Comment: In table2, the number of iterations is reported but it's not clear whether this is at test time or training time.*
>
> In Table 2, the number of iterations refer to the test time of a single model that was trained with four iterations for the camera and deformation estimation. We will revise our manuscript to make this subtle point clear.
>
> *Comment: I'm able to find a concrete description of how is correspondence trained. Is it fair to say the correspondence is trained by cycle consistency loss $L_{cycle}$?*
>
> Given that the correspondence drives the full mesh recovery process in our work, it is trained by the entirety from the self-supervised losses, with the cycle loss being one among them. In particular,
> we express the image-to-surface correspondences through the regressed 2D coordinates $\textbf{u}$ - where each 2D point is connected to a pre-determined 3D template point $\textbf{V}$.
> During back-propagation, these 2D coordinates $\textbf{u}$ receive  gradients from all self-supervised losses (e.g. mask loss, texture loss etc).
>
>
> *Comment: It is not clear how much deformation can the proposed approach handle ... such as articulation of human bodies.*
>
> We agree with the reviewer that a linear basis is insufficient for modeling large motion. For the case of human bodies or hands, a parametric model like SMPL/MANO may need to be used. There has been an approach towards that direction Kolotouros et al., however they require explicit 3D supervision for training, which is prohibitive for supporting any arbitrary object class. Recent research has focused on handling self-supervised deformations of categories that exhibit large motions [12], which paves the way towards self-supervised methods exhibiting large deformations.

---

### Official Review · Reviewer_t6xY · 2021-07-15

**Rating:** 6
**Confidence:** 4

**Summary:**

This paper tackles the problem of learning single-image 3D object reconstruction from single-view collections. The proposed method couples a correspondence prediction network with an iterative optimization procedure that solves shape and pose based on the predicted correspondences, starting with a category template shape.
The claim is that, with this iterative optimization procedure in the loop, it alleviates additional supervision for camera viewpoint such as keypoints, or multiple manual hypotheses.

The results show that this method is able to achieve comparable reconstruction accuracy without keypoints compared to previous methods that require keypoints. Overall, this paper presents an interesting approach of incorporating iterative optimization in a shape learning framework, but the method also has its limitations and the evaluation is rather limited.


**Ethical Concerns:**

I do not see immediate concerns.

**Limitations And Societal Impact:**

A few failure examples are provided in the supplementary, but they all seem very challenging cases, eg flying birds, heavy occlusion and small crops.
Are there failure cases on more regular images? Does the viewpoint prediction get stuck in local minima? Does the correspondence prediction fail dramatically, and what happens in the shape and pose optimization then?


**Main Review:**

## Strengths
### S1 - Method
The proposed approach is based on the same idea as in UMR[16] and ACSM[15]: separating the correspondence prediction and reconstruction, which is the key to classical structure-from-motion techniques. Unlike [15] and [16], the proposed method adopts an iterative optimization procedure in the reconstruction branch, whereas [15] and [16] use networks to predict shape and pose directly. Specifically, the camera is solved by L-BFGS optimization and the shape is solved in closed form with a optimized shape basis.

This has the advantage of being more robust against local minima given the shape pose ambiguity, which is a key challenge in unsupervised 3D reconstruction.
In contrast to existing methods that rely on additional keypoint supervision or multi-camera hypotheses, the proposed method is more data- and (perhaps) computation-efficient, although it still involves an iterative optimization process.

### S2 - Writing
The paper is well-written. The motivation is clearly introduced, and methodology clearly explained.


## Weaknesses
### W1 - Method
- W1.1: One major drawback of this optimization-based solution is that the same optimization procedure is also required at inference time, even though the authors show that even one iteration can already produce accurate results. How fast is each optimization iteration?
- W1.2: Does this optimization always yield a good solution? It seems to me the camera viewpoint can fall into local minima with BFGS optimization as well. Moreover, it relies heavily on the predicted correspondences, and cannot be robust against erroneous correspondences as a learning approach could with learned priors.
- W1.3: Another drawback is the way the correspondences are predicted using points. It does not guarantee continuity, as is also shown in the visualizations where the points can get messy in some parts. CSM [15] and part segmentation [16] have the advantage of producing smoother correspondences with an image. This also makes more concerned with the robustness of the optimization against errors in correspondence prediction.
- W1.4: One more major limitation is the need for a category template shape similar to [15] and [16]. This has largely simplified the problem of category-specific shape reconstruction and at the same time restricted the reconstructed shape from deviating too much from the template. The reconstructed airplanes for example are almost always the same despite the variation of the inputs (see Fig 9 in the sup. mat.).

### W2 - Evaluation
The evaluation has a number of limitations.
- W2.1: Why not compare with UMR[16]? It obtains part correspondences through a self-supervised part segmentation model, and does not require a template shape.
- W2.2: The numerical results are not very convincing. Mask mIoU only measures the shape from one single viewpoint, and does not evaluate the 3D shape. Keypoint reprojection is also very sparse and noisy, and may not be indicative of the 3D shape accuracy. It is unclear whether the comparison on these noisy metrics is actually meaningful.
- W2.3: The evaluation on Pascal3D+ with 3D mIoU is more meaningful (although the 3D shape is also "pseudo GT"), but the gap is relatively small. But more importantly, I wonder how well the template shape alone achieves in terms of metric?
- W2.4: Most of the visual results are visualized in the input viewpoint, which makes it hard to tell the how good the reconstructed 3D shapes are. From the only one novel view provided in Fig. 2 and 3 and in the sup. mat., the reconstructed 3D shapes do not look very plausible. The birds for example appear quite flat. The authors should provide videos or visualizations with multiple viewpoints to showcase the reconstructed 3D shapes. How do they compare to UMR and ACMR visually?
- It probably also makes sense to plot distribution of the predicted viewpoints. Is the model confused with some viewpoint, eg front vs back, left vs right?
- It could also help with understanding the pose and shape prediction to color the predicted mesh with the same color pattern in the template mesh.


## Clarification needed
- How exactly is texture symmetry enforced? The method samples texture from the input image using the predicted 2D points. In line 182, it briefly describes the texture symmetry enforced by averaging textures from two symmetric viewpoints. Does it mean taking the average of the sampled colors of two symmetric points (defined by the template)? If so, does this also rely on the visibility prediction? More details should be provided.
- This is interesting because texture symmetry alone can provide pretty strong cue for shape and pose. Does the model utilize this cue for reconstruction?
- There are 9 loss terms in total (if not more). How are they balanced?


## Additional comments
- Eq (3): $\mathbf{V}$ should be $\mathbf{T}$, input $\mathbf{C}$ should be lower case $\mathbf{c}$.
- Line 207: "vertical flipping"? ie, about the horizontal axis and the objects would be upside down?


---
## Post-Rebuttal
I appreciate the authors' efforts in the rebuttal, which has addressed many of my questions, including the computational cost, comparison with UMR, and template shape baseline performance on Pascal3D+. Regarding the robustness, it is still not clear how robust the correspondence prediction and subsequently the pose estimation are. This is hard to quantify, but more discussion on failure cases in the final version would be very helpful.

Overall, this paper has sufficient interesting material for acceptance. I will keep my original review as borderline accept.

**Time Spent Reviewing:**

5

---

> ### Author Response · Authors · 2021-08-10
> **Response to Reviewer's feedback**
>
> We summarize our differences with ACSM and UMR. First, ACSM requires multiple hypothesis, camera rotation constraints and manual part segmentation of the template to achieve articulated 3D reconstruction. Similarly, UMR relies on prior work (SCOPS) for part learning and multiple hypothesis. Further, UMR adopts an EM strategy for learning the template shape which is based on heuristics.  Instead, we directly perform 2D regression, delivering a leaner solution that relies on a single encoder and a single-stage optimization, instead of requiring  two-stage training or manual interventions.
>
>
> *Comment: W1.1: One major drawback of this optimization-based solution is that the same optimization procedure is also required at inference time, even though the authors show that even one iteration can already produce accurate results. How fast is each optimization iteration?*
>
> Each camera optimization iteration requires on average 34.7 msec and each deformation step 2.22msec, while using  unoptimized code, as detailed above. Please refer to relevant answer to Reviewer 1 above which thoroughly covers  computation concerns.
>
> *Comment: W1.2: Does this optimization always yield a good solution? It seems to me the camera viewpoint can fall into local minima with BFGS ... as a learning approach could with learned priors.*
>
> We do not claim that the optimization $\textbf{always}$ yields a good solution: it depends on the correspondences provided as input to NR-SFM. Even though it does not necessarily guarantee a good solution for every single image in the dataset, our method is a learning approach and back-propagates through the optimization scheme. From the point of view of optimizing the total objective of correct reprojection to the image and minimizing the self-supervised losses, our approach can be understood as block-coordinate descent combined with SGD.
>
> We agree that our method could further be improved by regressing the initialization of the optimization scheme, e.g. for the expansion coefficients on the basis. Still without it, and while using a leaner approach, we deliver clearly better results than methods that rely on CNNs entirely to deliver the pose and deformation estimates.
>
> Further, we mention that the "learned approaches" [10, 11, 12, 14, 15, 16] use hard-coded constraints in the camera prediction network for both the scale and the rotation of each axis, unlike our method which does not need any of these constraints. The "learned approaches" need these constraints to bootstrap the training process else the training procedure fails to learn a robust pose and deformation predictor.
>
> *Comment: W1.3: Another drawback is the way the correspondences are predicted using points ... This also makes more concerned with the robustness of the optimization against errors in correspondence prediction.*
>
> We do not agree that there is a clear drawback regarding accuracy; predicting smooth, UV-level representations like [15,16] does not necessarily guarantee that the  correspondences are good - they can be smooth (as outputs of CNNs) but wrong in their entirety as shown in the failure cases of the works [14,15,16]: to the same extent that a CNN can fail to predict correct 2D points, it can also fail to predict correct UV maps. Directly regressing points has the advantage of not requiring the localization of 2D points in images through their UV values, which is both time consuming and requires tricks to make differentiable.
>
> *Comment: W1.4: One more major limitation is the need for a category template shape similar to [15] and [16]. This has largely simplified the problem of category-specific shape reconstruction and at the same time restricted the reconstructed shape from deviating too much from the template.*
>
> We agree with the reviewer that this is a limitation of not only our method, but  all template shape methods [10, 11, 12, 14, 15, 16]. We see our work as a stepping stone to future work that can potentially accommodate topological flexibility, e.g. through SDFs to allow for larger deformations and examine complementary sources of supervision for the task of self-supervised monocular reconstruction. For instance Shelf-Supervised Mesh Prediction in the Wild, by Ye et al. reports exciting progress in this direction; we will cite and comment.
>
> *Comment: W2.1: Why not compare with UMR[16]? It obtains part correspondences through a self-supervised part segmentation model, and does not require a template shape.*
>
> Originally we had not compared with UMR[16] because the work reports metrics on Mask IoU and Keypoint Transfer which are not in line with the rest of the works. For the rebuttal, we have used the evaluation script of the publicly available code of UMR to evaluate our proposed method. The reported scores for all methods excluding ours are taken from [16].
>
> |  | CMR | CSM | UMR | UMR  w/o SCOPS | TTP | TTP |
> |---|:---:|:---:|:---:|:---:|:---:|:---:|
> | Keypoints | Y | N | N | N | Y | N |
> | Mask IoU | 0.706 | - | 0.734 | 0.744 | 0.760 | 0.752 |
> | KT (Camera) | 47.3 | - | 51.2 | 29.0 | 52.5 | 41.5 |
> | KT (Texture Flow) | 28.5 | 48.0 | 58.2 | 32.8 | 66.2 | 48.9 |
>
>
> As shown in Table 2, TTP with keypoints outperforms all competing method across the mIoU and Keypoint Transfer metrics. When there is no keypoint supervision, TTP retains its superiority for Mask IoU while for keypoints performance drops to 41.5 and 48.9 for KT Camera and Texture Flow, which is lower than SCOPS-based UMR. As also reported in  [16], performance drops significantly without a SCOPS-based loss. We note that SCOPS is a separate and orthogonal piece of work, related to 2D part discovery, which could be used in tandem with our approach to further improve keypoint scores. We will include these results and elaborate in our manuscript.
>
> *Comment: W2.2: The numerical results are not very convincing. ... It is unclear whether the comparison on these noisy metrics is actually meaningful.*
>
> We agree that the lack of 3D ground truth is a pain point for progress for in-the-wild monocular 3D category reconstruction overall.  Still, our quantitative results are in line with all prior works on self-supervised monocular 3D reconstruction [11, 12, 14, 15, 16] which report mask mIoU and keypoint reprojection. We believe that the combination of these two metrics and a large collection of qualitative visualizations provide insightful comparisons between methods.
>
> *Comment: W2.3: The evaluation on Pascal3D+ with 3D mIoU is more meaningful (although the 3D shape is also "pseudo GT"), but the gap is relatively small. But more importantly, I wonder how well the template shape alone achieves in terms of metric?*
>
> The 3D mIoU of the template of the car class is 0.5543 and 0.4142 for the aeroplane class. TTP's performance on these classes is 0.67 and 0.488   respectively which surpasses the performance of previous works on the Pascal3D+ dataset.
>
> *Comment: W2.4: Most of the visual results are visualized in the input viewpoint, which makes it hard to tell the how good the reconstructed 3D shapes are. ... How do they compare to UMR and ACMR visually?*
>
> We thank the reviewer for the recommendations to enhance our visualizations. Comparisons against UMR will definitely add value, beyond the comparison already existing against CMR and UCMR. Comparison against ACMR is impossible given that the code and pretrained models are not publicly available. Unfortunately we cannot provide these visualizations in the confines of the openreview format.
>
> *Comment: How exactly is texture symmetry enforced? The method samples texture from the input image using the predicted 2D points. ... Does the model utilize this cue for reconstruction?*
>
> The predicted texture of the invisible side is learned using a symmetric assumption which means that the depicted object exhibits textural symmetry across left-right. This allow us to render the invisible side of the mesh with a simple camera transformation and apply the same texture loss for the flipped rendering [10]. As R3 correctly suggests, our model uses the texture cue for reconstruction: indeed, as shown in Table 3a, removing the texture loss causes a drop in performance for both mIoU and PCK metrics which indicates that texture provides a cue for reconstruction.
>
> *Comment: There are 9 loss terms in total (if not more). How are they balanced?*
>
> Beyond the losses described in the paper, we do not have any other losses or hard-coded constraints. The weights of the losses are set according to their range in order to ensure that the sum of all losses will be close to 1 in the beginning of the training. We will share our code for reproducibility.
>
> *Comment: Line 207: "vertical flipping"? ie, about the horizontal axis and the objects would be upside down?*
>
> We mean left-right flipping and not upside down and we will fix it accordingly.

---

### Official Review · Reviewer_xnWE · 2021-07-16

**Rating:** 6
**Confidence:** 3

**Summary:**

The authors propose to separate out camera view point and template deformation using a differentiable optimization layer, and only delegate correspondence estimation to a CNN. The authors demonstrate improved performance on standard benchmarks with such a modular approach.

**Limitations And Societal Impact:**

Augmented reality is here and this work directly impacts it. The authors should include a broader discussion of negative impacts from an augmented reality perspective. Research in nascent stage is the best place to identify potential negative impact on society.

**Main Review:**

Strong points
- Formulating the camera viewpoint and template optimization as a differentiable layer and enabling end-to-end training
- Geometry driven losses requiring no annotation

Weak Points
- The experiments are limited. Especially, there is no experiment discussing the performance with/without the silhouette as is done in the related work,
- The improvement due to separating the optimization tasks should be better motivated and substantiated with additional experiments. For e.g., what happens if we were to use a cnn to regress the view point/deformation parameters with correspondence as input. How does it compare with the differentiable optimizer? What is the inflection point in performance wrt to number of iterations? As neural networks are general function approximaters, the burden of multi task optimization can be alleviated with more network capacity or additional data. All of these are design choices that are not discussed.
- The main idea of separating out using a CNN for correspondence and classical optimization for pose and shape regression is not new. Real-time Pose and Shape Reconstruction of Two Interacting Hands With a Single Depth Camera by Mueller et. al. were the first to do so to the best of my knowledge and there is related work in body/hand shape estimation which is not discussed in this paper. Learning to Reconstruct 3D Human Pose and Shape via Model-fitting in the Loop by Kolotouros et. al. is also relevant.
- No mention of run time or compute analysis.

Pose Rebuttal Review:
I apologize for missing the fact that mask is necessary for the approach. It was a detail overlooked by me while skimming related work as this is not my primary domain of research. Feeding correspondence to a network can be trivially accomplished by passing tuple of positions or their embeddings thereof. As demonstrated in transformers, this information can help the network learn better. I am still curios of such an analysis. I would also be more enthusiastic about an acceptance if the code is made available, as replication of the approach is non-trivial and marginal improvement over SOTA might be difficult to achieve in practice without the authors divulging the 'secret sauce' as common among the best of contributions in this domain. I am slightly upgrading the rating based on other reviews and their assessment along with the additional experiments on latency.

**Time Spent Reviewing:**

1.5

---

> ### Author Response · Authors · 2021-08-10
> **Response to Reviewer's feedback**
>
> We would like to thank the reviewer for reading our manuscript and providing feedback.
>
> *Comment: The experiments are limited. Especially, there is no experiment discussing the performance with/without the silhouette as is done in the related work.*
>
> We are not aware of  recent related works reporting results on unsupervised mesh recovery without silhouette supervision. We would have appreciated direct references to such works.
>
> Our experiments are inline with all prior works on self-supervised 3D reconstruction [9, 10, 12, 14, 15, 16, 28] and we provide thorough ablation studies that examine several aspects of the proposed method. In the closely related work to ours [9, 10, 12, 14, 15, 16, 28] there hasn't been any experiment with and without silhouette because it is impossible to train any of these approaches without masks. The silhouettes are the main component of all these methods in order to learn the pose and deformation in a self-supervised manner.
>
>
> *Comment: The improvement due to separating the optimization tasks ... compare with the differentiable optimizer?*
>
> It is unclear to us how correspondences could be used as input to a CNN. There is no supervision signal for correspondence - as such it would need to be "discovered" in an unsupervised way. If it were to be obtained from the image, by the data processing inequality, this means we provide less info than raw image - making the CNN's predictions weaker than the plain RGB-driven baseline. In our eyes correspondence is only useful in terms of simplifying the 3D reconstruction problem.
>
>
> *Comment: What is the inflection point in performance wrt to number of iterations?*
>
> We present in Table 2 the change in performance with respect to the number of iterations. The performance stagnates after 3 iterations and further computation yields minimal improvements.
>
> *Comment: As neural networks are general function approximaters, the burden of multi task optimization can be alleviated with more network capacity or additional data.*
>
> We agree that in principle a CNN could directly learn the 2D-to-3D mappings with just a CNN regressor - in practice this would only be possible in the presence of abundant 3D supervision and strong priors (e.g. this has been done for face and body mesh reconstruction).
> In our minimally supervised case (using only mask supervision) we are handling an perversely ill-posed problem. Our method is in our understanding among the leanest for delivering monocular mesh reconstruction, while all existing works use multi-tasking (e.g. CSM and ACSM use two separate networks to achieve this, one for the correspondences and the other for image encoding)
>
> CMR is a purely deep-learning method that is closest  to ours, relying on the same image encoder, i.e. a ResNet18. As shown in Table 1 of the manuscript, we outperform CMR by a significant margin. We attribute this to the explicit treatment of geometric correspondence through NRSFM.
>
> Finally, we used all available data for training while being in line with all relevant work [9, 10, 12, 14, 15, 16, 28] to ensure fair quantitative comparisons. We agree that more data can only improve performance, but without the right inductive priors it seems to us unlikely to have rapid progress in unsupervised 3D mesh recovery.
>
> *Comment: The main idea of separating out using a CNN for ... in the Loop by Kolotouros et. al. is also relevant.*
>
> We provide thorough references to prior works on NR-SFM [7,8,32,33] and deep NR-SFM [13,34,35] which are closely related to our work. Please note that our focus is on generic category reconstruction, and not humans in particular; as such we cannot rely on  strong parametric priors, like SMPL/MANO,  pipelines for synthetic data, or depth sensors, as in the  references indicated by R1 (xnWE). All we have as inputs are 2D images and segmentation masks.
> As such, we face the critical challenge of disentangling 3D camera pose from 3D non-rigid shape through unsupervised learning (as in NR-SFM), which is not faced in the indicated works.
>
> We do cite the work of Kolotouros et al (lines 365-367) [21] and also provide extensive references to other relevant works that span in the area of body/hand shape estimation [1, 19, 20, 21, 22, 23, 30, 39]. We will also add the  Mueller et. al. reference but we think it is out of scope compared to our task, based on our argument above:  we agree that there is overlap on the  general concept of establishing correspondences for 3D reconstruction,  but as R1 (xnWE) also mentions, this is as old as 3D vision itself, and has also been used in the context of deep learning.
>
> *Comment: No mention of run time or compute analysis.*
>
> In the supplementary material (lines 514-517) we already provided a short run time analysis of our method, but appreciate it should appear in the main submission. For the rebuttal, we extend the run time analysis further to address relevant computational concerns.
>
> |  | Device | Iters | CNN (msec) | Optimization (msec) | Total (msec) |
> |---|:---:|---|:---:|:---:|:---:|
> | CMR | GPU | - | 5.11 | N/A | 5.11 |
> | CSM | GPU | - | 150.73 | N/A | 150.73 |
> | ACSM | GPU | - | 191.46 | N/A | 191.46 |
> | TTP | GPU | 1 | 4.02 | 33.4 | 37.46 |
> |  |  | 4 | 4.02 | 125.14 | 129.16 |
> | TTP | CPU | 1 | 18.52 | 33.44 | 51.96 |
> |  |  | 4 | 18.52 | 125.14 | 143.67 |
>
> In Table 1 we provide a run time analysis of our and prior methods on the task of self-supervised 3D reconstruction. The analysis was performed on a machine with a NVidia RTX 2080Ti, an Intel Xeon W-2255 and 32 GBs of RAM. We report the average of 20 runs for the reconstruction of a 256x256 image.
>
> To ensure that the benchmark is fair for all methods, we compute the execution time between the moment an image is given as input to a network up to the moment where the network predicts the mesh, the camera pose and the texture; the implementations of the methods used for comparison are the publicly available ones from the original authors.
>
>
> As reported in Table 1, TTP is faster than ACSM by a considerable margin even on the CPU. CMR is the fastest method of all due to its simplicity, however,  it requires keypoint supervision and has substantially lower results as indicate in Table 1 of our manuscript.
>
> We note that our implementation relies on PyTorch and we have used PyTorch's python-based LBFGS optimizer for convenience and autograd for Jacobian computation; understandably for AR applications the LBFGS timing can become substantially faster in C and with explicitly coded Jacobian matrix computation.
>
> *Comment: Augmented reality is here and this ... identify potential negative impact on society.*
>
> We agree that our work can have a substantial impact on AR, by allowing us to integrate deformable 3D objects in interactive experiences and appreciate R1's concerns regarding potential impacts. We anticipate AR as  a technological breakthrough, comparable only to the impact of smartphones and the internet to our lives.
> Still, we believe that our work's implications are neutral, since we focus on unsupervised learning, and its long-term implications can amount to integrating into AR inanimate objects (cars, bikes etc) or animals (birds, cows, etc) as first-class citizens. We will elaborate.

---

### Decision · Program_Chairs · 2021-09-27

**Decision:**

Accept (Poster)

**Comment:**

The paper presented a new, iterative optimization procedure in the reconstruction branch of the conventional 3D reconstruction framework, instead of training network s to predict shape and pose directly as in UMR and ACSM. In general, the reviewers were a bit mixed in the initial review. After the rebuttal and the discussion phase, R1 increased the rating. Please include many of the rebuttal points in the final version, especially run time analysis and comparison with UMR.